# Automated Real-Time Eddy Current Array Inspection of Nuclear Assets

**DOI:** 10.3390/s22166036

**Published:** 2022-08-12

**Authors:** Euan Alexander Foster, Gary Bolton, Robert Bernard, Martin McInnes, Shaun McKnight, Ewan Nicolson, Charalampos Loukas, Momchil Vasilev, Dave Lines, Ehsan Mohseni, Anthony Gachagan, Gareth Pierce, Charles N. Macleod

**Affiliations:** 1SEARCH: Sensor Enabled Automation, Robotics & Control Hub, Centre for Ultrasonic Engineering (CUE), Department of Electronic & Electrical Engineering, University of Strathclyde, Royal College Building, 204 George Street, Glasgow G1 1XW, UK; 2National Nuclear Laboratory LTD., Warrington WA3 6AE, UK; 3Sellafield LTD., Sellafield, Seascale, Cumbria CA20 1PG, UK

**Keywords:** non-destructive evaluation, robotic NDE, automated eddy current testing, eddy current arrays

## Abstract

Inspection of components with surface discontinuities is an area that volumetric Non-Destructive Testing (NDT) methods, such as ultrasonic and radiographic, struggle in detection and characterisation. This coupled with the industrial desire to detect surface-breaking defects of components at the point of manufacture and/or maintenance, to increase design lifetime and further embed sustainability in their business models, is driving the increased adoption of Eddy Current Testing (ECT). Moreover, as businesses move toward Industry 4.0, demand for robotic delivery of NDT has grown. In this work, the authors present the novel implementation and use of a flexible robotic cell to deliver an eddy current array to inspect stress corrosion cracking on a nuclear canister made from 1.4404 stainless steel. Three 180-degree scans at different heights on one side of the canister were performed, and the acquired impedance data were vertically stitched together to show the full extent of the cracking. Axial and transversal datasets, corresponding to the transmit/receive coil configurations of the array elements, were simultaneously acquired at transmission frequencies 250, 300, 400, and 450 kHz and allowed for the generation of several impedance C-scan images. The variation in the lift-off of the eddy current array was innovatively minimised through the use of a force–torque sensor, a padded flexible ECT array and a PI control system. Through the use of bespoke software, the impedance data were logged in real-time (≤7 ms), displayed to the user, saved to a binary file, and flexibly post-processed via phase-rotation and mixing of the impedance data of different frequency and coil configuration channels. Phase rotation alone demonstrated an average increase in Signal to Noise Ratio (SNR) of 4.53 decibels across all datasets acquired, while a selective sum and average mixing technique was shown to increase the SNR by an average of 1.19 decibels. The results show how robotic delivery of eddy current arrays, and innovative post-processing, can allow for repeatable and flexible surface inspection, suitable for the challenges faced in many quality-focused industries.

## 1. Introduction

The global Non-Destructive Testing (NDT) market size was valued at USD 6.3 billion in 2021 with a predicted compound annual growth rate (CAGR) of 13.66% from 2022–2029 to hit a total market value of USD 16.66 billion [1]. This high level of growth can be attributed to the rise of “NDT 4.0”, in which greater connectivity across the manufacturing supply chain is sought through the integration of connected sensors of which NDT techniques play a role [2]. To deliver this level of interconnectivity, it is now commonplace to see automated robotic delivery of NDT [3,4,5,6].

The vast majority of the NDT market is based on volumetric inspection of high-value infrastructure and components, such as automotive/aerospace components or public rail infrastructure, primarily through the use of radiographic and ultrasonic testing. Due to this popularity, the automation of volumetric techniques is the most mature in the NDT industry. Further growth in the automation of volumetric NDT is expected to lag behind other NDT techniques, as innovation has shifted towards more novel and complex delivery of volumetric NDT as well as incorporating advanced imaging and post-processing techniques. Examples of these trends include performing the volumetric inspection at the point of manufacture for high-value components [7,8,9,10,11,12], performing aerial UAV-based volumetric inspection [13,14,15,16], optimising the amount of data gathered [17,18], and deploying machine/deep learning in the analysis of the datasets generated [19,20,21].

By contrast, the automation of surface inspection is far less mature and from 2022–2029 it is predicted to have the highest CAGR of any NDT technique due to the increased adoption of Eddy Current Testing (ECT) [1]. Of the ‘big 5′ NDT techniques, eddy current, magnetic particle, and penetrant testing were shown to be able to detect surface-breaking flaws, where others in the ‘big 5′ (ultrasound and radiographic) struggle [22].

Eddy currents are induced in a sample according to Faraday’s Law of Induction [23] when a coil carrying an alternating current produces an alternating magnetic field and the conductive sample lies within this magnetic field. The induced eddy current in the sample is of the opposite phase to that of the coil and sets up its own magnetic field to oppose that of the coil. The eddy current density, J(z), decays exponentially with depth z in an isotropic material, and the sensed impedance is directly proportional to the current density [24]:(1)$$J\left(z\right)=J_{0}\exp\left(-\frac{z}{\delta_{sd}\left(1+i\right)}\right)\;$$

In the presence of a defect, the current density is altered and this change in current density can be sensed as a change in impedance. The magnitude of the eddy current density decays exponentially and when it falls to 1/e of its surface value, the depth at which this occurs is known as the standard depth of penetration, $\delta_{sd}$. The standard depth of penetration is dependent on the frequency of the voltage in the coil, the magnetic permeability, and the electrical conductivity of the component, and is widely viewed as the deepest depth a meaningful change in impedance can be sensed. Due to the exponential decay associated with eddy currents, they are ideally suited to detecting surface-breaking defects. This is in direct contrast with ultrasound where the front wall echo typically masks any shallow defects within a component. With correct eddy current probe design and frequency selection, an eddy current can be created that has a standard depth of penetration greater than or equal to the thickness of some thin-walled components, such as the canisters used in the storage of low-level nuclear waste.

Magnetic particle testing is restricted to the use of ferromagnetic metals and requires the component to be magnetized/de-magnetized frequently. While penetrant testing is not restricted to any material but requires the component to be coated in a penetrant and developer, which is frequently undesirable. Both magnetic particle and penetrant testing are subject to great operator error and do not produce discrete data points as a sensor is rastered across the component’s surface making automation unfeasible. However, these drawbacks do not exist for ECT, and hence ECT is well suited for automation. As society moves towards Industry 4.0, automation is becoming increasingly important in surface inspection in the immediate future.

In comparison to volumetric techniques, ECT does not suffer from the health and safety concerns associated with radiographic inspection. Additional technical requirements may also prohibit the use of other inspection modalities. For example, multi-angle accessibility requirements and part size limitations may make computed tomography radiographic testing unfeasible. While for ultrasonic inspection, environmental factors may deter the use of a couplant. ECT has a significant advantage as single-sided access is all that is required, and no couplant is needed to perform an inspection.

Reuse and sustainable business practices are the main drivers behind the increased adoption of ECT, as detecting surface-breaking flaws that occur in operation is becoming increasingly important to prolong the safe operation of key assets for industries such as nuclear and aerospace. Furthermore, due to the lower market size, robotic delivery of ECT is far less common with only a few primitive integration efforts being reported in the literature [25,26,27,28]. To keep pace with the high throughput demands of modern production/maintenance lines, increased robotic deployment of ECT is necessary and vital to capitalise on this demand.

This paper presents, for the first time, the automated deployment of an eddy current array, via a flexible robotic cell complete with force–torque control, to scan a canister typical of the ones used in the storage of spent nuclear fuel. Table 1 shows a comparison between previously published papers that feature robotically deployed eddy current inspection and this work. Real-time adaptive control of a 6-axis robotic arm (KUKA Quantec Extra HA KR-90 R3100, Augsburg, Bavaria, Germany [29]) and an external rotary stage (KUKA DPK-400 [30]) with force–torque compensation was accomplished using a framework described in the author’s previous work [31]. Force–torque compensation allowed for constant lift-off of the eddy current array during the inspection. This was intentionally carried out as it was shown that robotically delivered eddy current inspection offers far less noise when compared to that of manual eddy current inspection [32]. A commercial 32-element padded eddy current array from EddyFi (Part No: ECA-PDD-034-500-032-N03S, Québec, QC, Canada [33]) with a centre frequency of 500 kHz and an operating frequency range of 100–800 kHz, along with a 64-element Eddyfi Ectane 2 controller [34] were used to perform 180-degree rotary scans of a 1.4404 stainless steel nuclear grade canister with known stress corrosion cracks. Extensive software infrastructure coupled with the Eddyfi Ectane 2 Software Development Kit (SDK) allowed for the impedance data to be logged and analysed in real-time. All data were stored in a proprietary binary file format to allow for further post-processing in MATLAB.

This infrastructure allowed for the acquisition and real-time analysis of impedance data. Novel image post-processing techniques, such as phase rotation and mixing, were shown to increase the Signal to Noise Ratio (SNR) of the resulting C-scan images by an average of 4.56 and 1.19 decibels, respectively. It is envisaged that studies such as this will progress eddy current testing to match the level of flexibility and quality enjoyed in the post-processing of ultrasonic datasets [35,36].

## 2. Experimental System

NDT is crucial to safety-conscious industries such as nuclear [37]. Traditionally, the inspection of nuclear assets is highly resource-intensive and complex. The inspection challenge is complicated further when the asset lifetime exceeds the original design intent. This problem is one that is currently being faced in the UK, where government policy has shifted from favouring reprocessing to long-term storage of nuclear assets [38]. Spent nuclear fuel due for reprocessing is now being stored long term as reprocessing facilities are closed down. Some sites store low-level waste in canisters made from 0.9 mm thick 1.4404 stainless steel. These canisters range from 130–150 mm in diameter and are ~300 mm in length. To allow for effective cooling, the canisters are stored in facilities that are partially open to the environment. Given the coastal location of the UK, stress corrosion cracking is a concern. Due to the points mentioned above, canisters with intentionally induced stress corrosion cracks were scanned with an eddy current array and the acquired impedance data were analysed within this study.

### 2.1. Hardware and Experimental Summary

Figure 1 shows the experimental hardware used in the automated deployment of the eddy current system. A nuclear canister with a matrix of 16 stress corrosion cracks shown in Figure 2 is held within a mechanical chuck on top of a KUKA DPK-400 external rotary stage that has an angular resolution of 0.009°. The padded Eddyfi eddy current array (Part No: ECA-PDD-034-500-032-N03S) is mounted in a bespoke 3D-printed housing which is in turn secured to an IP-65 rated gamma force–torque sensor from ATI Industrial Automation (Apex, NC, USA) [39]. To move the sensor to the height of interest for the inspection, the eddy current array, 3D-printed housing and force–torque sensor assembly, are mounted to the flange of a KUKA KR-90 robot. Both the KR-90 and DPK-400 external rotary stages are controlled via a KRC 4 controller [40].

In order for the eddy current array to be pressed onto the canister surface in the direction of the canister’ centre, a calibration tool was manufactured to teach the KR-90 robot a new base coordinate system. The calibration tool was made so that it would align the centre of the chuck to the centre of the rotary stage. Additionally, the calibration tool allowed for the centre of the tool along with 4 concentric radial calibration points at 150 mm in 90° increments to be taught to the KR-90 robot. By teaching the KR-90 robot these points, it was able to know where the centre of the canister and rotary stage was relative to its own coordinate system, and ensure motion was performed relative to this point. This effort guaranteed that the eddy current array was always pressed against the canister surface in the direction of the canister’s axial centre and helped establish good electromagnetic coupling during the automated inspection.

The eddy current array is deployed to the height of interest in the Z-direction of the canister via a variable set by the user on the Graphical User Interface (GUI) of a LabVIEW program using a framework similar to previously published work [31]. Force and torque in and around all three axes shown in Figure 1 are sensed via the force–torque sensor and are transmitted to a LabVIEW program via the robot controller using the Kuka Robot Sensor Interface (RSI) [41]. The transmission of the force and torque characteristics allowed for: (1) the adaptive motion of the eddy current sensor during inspection; (2) the balancing of the eddy current probe and the subsequent triggering for the acquisition of the impedance data to begin; and (3) the triggering of the rotary stage to begin movement. It is important to note that the force–torque sensor was calibrated with all hardware mounted prior to any automated inspection through a program provided by the manufacturer. The calibration enabled the net force and torque values being applied to the eddy current array and mounting assembly to be correctly sensed and subsequently transmitted to the LabVIEW control program for adaptive motion to be performed.

The KR-90 robot presented the eddy current array onto the surface of the nuclear canister at the user-specified height, and a target force and torque of 10 N and 0 Nm were met in the Y-direction and X-axis, respectively, for 3 s. Once this time period had passed, the balancing of all coils within the eddy current array was performed when the probe was stationary. After a further 3 s, the impedance data acquisition along with the rotary stage movement was triggered.

During the inspection, a PI control system was used to monitor and correct both the force in the Y-direction and the torque around the X-axis at the previously mentioned target force and torque values. It was found that ***P***- and ***I***-values of 0.1 and 0.0 gave an adequate control response. Control of the eddy current probe’s orientation in this manner allowed for minimal variations in the lift-off of the eddy current array throughout the inspection providing excellent coupling. Other previously published literature has shown that lift-off can be reduced via advanced signal processing and elaborate probe design [42]. These efforts are often particularly involved and particular to one sample/defect type. As a result, these lift-off compensation strategies are complex to deploy and benefit from. The approach in this paper of utilising a force–torque sensor in combination with a padded ECT array provides experimental flexibility and passively compensates for any lift-off variation at the point of acquisition giving wide-reaching benefits.

The acquisition of the impedance data was stopped when the rotary stage had completed the angular movement requested by the user from within the LabVIEW program. A singular scan can be summarised by the following process:A connection with the eddy current Ectane device is made.The eddy current array is set up with the following parameters:
Probe type;Probe configuration (axial and/or transversal—See Section 2.2);Frequencies;Voltages;Gain;Repetition rate.The robot and external rotary axis are set up with the following parameters:
Linear speed of the KR-90 robot;Approach speed of the KR-90 robot;Angular movement of the canister/external rotary stage;Angular speed of the canister/external rotary stage;Target force for the KR-90 robot to apply the array onto the canister.The KR-90 robot places the probe against the canister and the target force is reached.The target force is maintained for 3 s.The balancing of the eddy current array is performed.Wait a further 3 s.The acquisition of impedance data and rotary stage movement is triggered.Once full angular motion is complete, the acquisition of impedance data is stopped.The KR-90 robot moves the eddy current array to a predetermined safe position.The acquired impedance data are saved to a binary file for post-processing in MATLAB.

### 2.2. Eddy Current C-Scan Acquisition

Figure 3 shows a generic eddy current array layout along with illustrations of the transmit and receive pairings for the axial and transversal configurations. Depending on the probe geometry, there may or may not be an equal number of transversal and axial transmit and receive pairs. Each pairing in each configuration generates a data point of complex impedance data. The probe is linearly scanned perpendicular to the coil columns as noted in Figure 3, and the data points are logged into a complex 2D array. The resulting complex arrays can then be post-processed, and the vertical component of the post-processed complex array can be plotted in a C-scan format to show any defective signals with maximum Signal to Noise Ratio (SNR).

As can be seen in Figure 3b for the axial configuration, coils in the array are excited in one column and reception of the impedance data is performed across the array in the second column. Conversely, the transversal configuration documented in Figure 3c shows coils being excited and reception of the impedance data being performed within the same vertical column of coils.

The coil firing sequence is changed between the axial and transversal configurations to achieve greater sensitivity to differing defect orientations. With reference to the coordinate system in Figure 3, a larger change in impedance would be observed for a defect that is aligned with the X-axis for a transversal configuration over that of an axial configuration. This is due to the defect more severely intercepting the eddy current that exists between the two transmit and receive coils in the transversal configuration over that of the axial configuration. This greater compression of the eddy currents caused by the defect presence will have a large effect on the electromagnetic field and by proxy the sensed change in impedance. The opposite can be said to be true for a defect aligned in the Y-direction. For further reading, Ye et al. [42] provide a thorough theoretical and experimental investigation of this phenomenon.

It is also evident from Figure 3 that the centres of excitation are not aligned between the axial and transversal datasets in the X-direction. Moreover, for each coil column within the transversal dataset, the data centres are also misaligned. As alluded to in Section 2.2, this positional misalignment is corrected within the LabVIEW program and ensures that the resulting complex array for each dataset has the same spatial grid.

Key to the positional compensation is the acquisition rate of the eddy current array and the angular speed of the rotary stage so that each acquisition point aligns with an integer number of divisions of half the array coil column pitch, Δx. The acquisition rate and number of divisions between half of the array column pitch are set by the user, and the coil pitch is defined by the geometry of the array. These three variables are used to set the angular speed of the rotary stage. For example, if an eddy current array has a column coil pitch of Δx=7 mm, an acquisition rate of 50 Hz, and 50 divisions, the linear speed would need to be $\left(\frac{7}{2}\times \frac{1}{50}\right)/\left(\frac{1}{50\;}\right)=3.5\;\mathrm{mm}/s$. This linear speed can then be converted to rotational speed by dividing the diameter of the canister at 150 mm to give the angular speed of the rotary stage at 1.34 deg/s. Whilst individual datapoints are not positionally-encoded, the positional location is extrapolated from setting the angular speed relative to the eddy current probe geometry and acquisition rate as mentioned above. By doing so, it ensures that data are acquired at both the axial and transversal data centre points on the X-axis as the array is linearly scanned.

In order to ensure a common spatial grid, the first and last impedance data points corresponding to a distance of half the coil pitch are discarded within the axial complex array. By discarding the first set of data points that cover half the coil pitch, the axial complex array in the X-direction is synched with the first/odd column of the transversal dataset. Moreover, by discarding the last set of data points that cover half the coil pitch, the axial complex array in the X-direction is synched with the second/even column of the transversal complex array. This discarding of data is shown graphically in Figure 4a. The resulting data is then linearly interpolated in the Y-direction to align with the Y-coordinates of the transversal complex array.

The transversal C-scan array is similarly compensated by separating out the first/odd and second/even columns into separate arrays. Impedance data corresponding to a distance of a full coil pitch is discarded from the start of the odd array. Conversely, the opposite operation is performed on the even array where impedance data corresponding to a distance of a full coil pitch is discarded from the end of the array. This process is graphically illustrated in Figure 4b. Once all data are discarded, the odd and even arrays are interleaved together to make one C-scan array that is on the same positional grid as the axial C-scan array.

Once all data were collected and positionally compensated, oversampling is undertaken in the vertical direction of the array. No oversampling is performed in the horizontal scan direction as this is controlled adequately by setting the rotational speed and acquisition rate of the robot as described in the previous paragraphs. The oversampling is performed via linear interpolation of the raw impedance data. It was found that this linear interpolation was fast to implement and produced negligible errors with a maximum error of 2.12% and an average of 0.55% across both the axial and transversal datasets at 250 kHz.

By performing data compensation in this manner, a common spatial grid is established for each dataset configuration, enabling like-for-like comparison and further advanced post-processing techniques such as mixing of datasets.

### 2.3. Software Infrastructure

Extensive software infrastructure to control the eddy current Ectane device, as well as receive and process the acquired impedance data in real-time was developed and is documented in Figure 5. Literature has previously well documented the robotic software infrastructure required [31,43] and as a result, the work presented herein will focus on the eddy current software development effort.

In total 3 programs were developed: (1) A C program that houses the Eddyfi Ectane 2 SDK; (2) A LabVIEW program that receives, post-processes and plots impedance data in real-time as well as saving the data in a binary file format; and (3) A MATLAB reviewer program that reads in the binary file for further post-processing.

Both the C and LabVIEW programs are state machines. States within the C program are evaluated through a switch statement within the main while loop. In addition to the main while loop, the C program contains two threads that each have local host Transmission Control Protocol (TCP) connections. The first listens for standardised comma-separated string commands from LabVIEW and the other sends 32-bit impedance data from the Ectane device to the LabVIEW program. The same infrastructure with reverse logic is mimicked within the LabVIEW program through JKI state machines [44]. The standardised comma-separated string that is sent from LabVIEW is carried out in the following format:



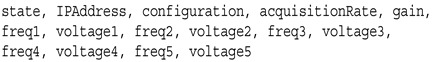



As can be seen, there are 15 variables housed within the standardised string command. The first of which is the state that the C program should execute, and these are summarised below.

Do Nothing;Connect to Device;Set Up;Balance;Acquire Data;Stop Data;Disconnect from Device.

The second is the IP address of the Ectane device in order for the C program to connect to the Ectane device. Third is the configuration of the probe (i.e., will axial and/or transversal datasets be acquired? What probe is being used?). Next is the acquisition rate and gain of all Ectane channels. The final ten are the voltages and frequencies of each Ectane channel. As the Ectane device can acquire 5 datasets at different voltages and frequencies each of these must be specified even if some are unused.

The raw impedance data are received in the LabVIEW program as a series of 32-bit numbers and are immediately queued to be sequentially analysed in two additional threads. Using a 6 core, 2.6 GHz Intel i7-8850 H processor, it was found that the queueing of the received data was performed in 1 ms. As previously, these threads are implemented via two JKI state machines.

The first thread takes each 32-bit number and separates out the first and last 16-bits of data as these correspond to the imaginary and real impedance components. Additionally, the first thread reformats the impedance data into geometric order as the coils are pulsed in a pseudo-random fashion to prevent crosstalk caused by mutual inductance. Moreover, the first thread compensates for offset in coil excitation in the X-direction. Further details of this coil excitation compensation are provided in Section 2.2. It was found that this process was executed in 1 ms on a 6 core, 2.6 GHz Intel i7-8850H processor.

The second thread within the LabVIEW program is responsible for interpolation in the Y-direction between axial and transversal dataset configurations, oversampling, basic mixing of different datasets and live plotting of the impedance magnitude. As before, further details of this Y-direction interpolation and mixing of datasets are provided in Section 2.2 and Section 2.4, respectively. Likewise, it was found that this process was executed in 5 ms on a 6 core, 2.6 GHz Intel i7-8850H processor. It is noted that the timings reported should be representative of any array used as the software infrastructure is built for the maximum number of elements, channel pairings, and number of frequencies.

This multi-threaded approach is illustrated in Figure 6 and provides data acquisition, positional compensation, and interpolation of impedance data whilst displaying various impedance magnitude C-scans in real-time to the user, all within the LabVIEW software environment with minimal 7 ms lag. The user can then select a directory to store the acquired data in a binary file format for future post-processing and analysis.

### 2.4. Image Enhancement of Impedance Data

It was shown in the literature that the impedance plane of the acquired data can be complex to interpret and variations in probe lift-off and wobble can commonly be mistaken as signals from defects [45,46]. Therefore, great care was taken in this work to minimise these adverse effects. Methods such as optimal probe design [47], multi-frequency excitation [45], and phase rotation [46] were shown to reduce such effects. Due to this work utilising commercial off-the-shelf (COTS) equipment, only multi-frequency excitation and phase rotation were performed. Multi-frequency excitation of 4 separate frequencies was conducted as the data were acquired and mixing of the datasets as described in Section 2.4.2 was performed in post-processing. Additionally, phase rotation was performed on the acquired C-scan datasets. All post-processing was performed via the MATLAB review application mentioned in Section 2.3.

#### 2.4.1. Phase Rotation

The signature of adverse effects such as lift-off and wobble experience a phase difference in the response caused by a defect on the impedance plane. It is therefore common to phase rotate the data so that the response from the lift-off aligns with the horizontal axis of the display impedance plane, and plot C-scan images of the resulting vertical component of the impedance [48]. Due to the phase difference observed between the lift-off variations and that of a defect, the resulting C-scan will show any response from a defect clearly.

Mathematically, this is described in Equations (2) and (3). Equation (2) describes the resulting acquired impedance array from Section 2.3, and Equation (3) describes the mathematical operation performed to phase rotate the data by an angle, θ. This can be carried out at the point of acquisition or in post-processing. For this study, the decision was taken to phase rotate the data in post-processing to maintain maximum flexibility with the acquired data.
(2)Z=R+iX
(3)$$Z_{rot}=Z(\cos\left(\theta \right)+isin\left(\theta \right))=\left(R+iX\right)(\cos\left(\theta \right)+isin\left(\theta \right))$$

#### 2.4.2. Mixing Eddy Current Datasets

As the impedance data were acquired onto a common spatial grid, mixing of datasets recorded under differing configurations or frequencies can be performed by superimposing the impedance C-scan data. This is graphically illustrated in Figure 7.

Two differing mixing methodologies were performed with the first being a simple sum and the second being a selective sum and average. As the name implies, the simple sum summates complex impedance datasets on a pixel-wise basis. For the selective sum and average, data above a defined noise floor were summated and everything below was averaged. The noise floor was defined as being 5 times the RMS values reported across a non-defective section of one of the impedance datasets to be mixed.

## 3. Results

Three 180-degree scans of the canister shown in Figure 2 were undertaken with both transversal and axial datasets being simultaneously acquired at frequencies of 250, 300, 400, and 450 kHz with an amplitude of 2 volts for each frequency channel, 30 dB of gain, an acquisition rate of 40 Hz, and a rotational speed of 1.72 deg/s. Each scan covered an area of 7687.1 mm^2^ (array height of 32.625 mm × half the circumference of a 150 mm canister equating to 235.62 mm) making the final stitched image representative of an area of 23,061.3 mm^2^. The interpolation was set to five, and the increments between half a coil pitch were specified at 20, giving a spatial resolution of 0.225 mm and 0.0563 mm in the vertical and horizontal directions, respectively. Positions were chosen for each scan so that they were acquired one array coil above each other with no overlap. The impedance data for all three scans were vertically stitched together and axial channel C-scans of the vertical impedance component from the impedance vector are shown in Figure 8. One of the stress corrosion cracks in the centre of the far-right column is highlighted. To the right of each C-scan, the impedance plane Lissajous for the highlighted defect is also shown along a horizontal cursor passing through the maximum intensity of the defect indication in the C-scan. It can be seen, that the impedance plane response of the same defect for different frequencies varies drastically in amplitude and phase due to the differing interaction depth of the eddy currents with the defect [46].

Additionally, Figure 8 also shows that at 250 kHz and 450 kHz, the impedance plane contains a large horizontal component and as such the resulting image contains a large amount of noise. In order to compensate for this effect, the impedance data at each frequency were phase rotated so that the SNR of the highlighted defect was maximized – see Figure 9.

Figure 10 shows C-scan images of the optimised phase rotated axial data, while Table 2 denotes the SNR increases for both axial and transversal datasets at all frequencies recorded for the target defect. The increase in SNR for all defects is visually evident in Figure 10, and on average, the SNR was increased by 4.56 decibels for the targeted defect. This result illustrates the effectiveness that phase rotation can have on increasing the image performance of C-scans and the benefit of being able to flexibly perform such a task in post-processing.

To further enhance image quality and reveal more about the nature of the defect, a mixing of different datasets, as described in Section 2.4.1, was performed. The optimised transversal and axial datasets at 250 and 450 kHz were mixed together, as the dissimilar frequencies would produce differing eddy current penetration depths and thus be influenced in differing manners. Equation (4) mathematically describes the penetration depth of an eddy current for a given material, where *f* is the frequency of the voltage being excited in the array coils in hertz (Hz), μ is the magnetic permeability of the component under test in henries per meter (H/m), and σ is the electrical conductivity of the component under test in siemens per meter (S/m).
(4)$$\delta =\frac{1}{\sqrt{\pi f\mu \sigma }}$$

For stainless steel, with an electrical conductivity of 1.08 × 106 S/m, and a relative magnetic permeability of 1.0025, a frequency of 250 kHz would produce a penetration depth of 0.967 mm, while a frequency of 450 kHz would produce a penetration depth of 0.721 mm.

The resulting mixed C-scan image is shown in Figure 11. Table 3 documents the SNR of the highlighted defect. As is shown in Table 3, the SNR of the defect for the simple sum approximates to be the average across all four datasets that contributed to the mixed image, and as such it can be said the imaging performance has not been improved by this mixing methodology. Interestingly, this is a result that is also observed in ultrasound when fusing multi-modal Total Focused Method (TFM) images [49]. By contrast, the selective sum and average technique were able to boost the SNR by an average of 1.19 dB, demonstrating an increase in imaging performance.

It is acknowledged that in this study, SNR is the only metric being used to evaluate the eddy current detection system. A better metric would be a physical parameter related to the geometry of the defect itself (i.e., crack extent, crack depth) and whether this is better reflected in the mixing of datasets. As reported in the literature, this is a highly complex inversion problem, with successful inversions demonstrated on only simple geometries [50,51,52,53] or overall dimensions such as the depth or extent on complex defect geometries [54,55]. In all these studies, the defects were manufactured to specified geometries before eddy current testing which is somewhat removed from a real inspection scenario where prior knowledge of the defect geometry is not known. In addition, the sizing algorithms used vary drastically from defect to defect making the inversion of defect size somewhat deterministic and not well suited to automated deployment and analysis with which this paper is concerned. While the current system and signal processing cannot currently invert physical defect size, it was shown on another sample and different probe that is better suited to low-frequency operation, that the system is able to detect embedded defects ~3 mm below the inspection surface.

To understand more about the physical geometry of the highlighted stress corrosion crack, a macrograph was taken at 96 times zoom and is shown in Figure 12. It can be seen that the defect under inspection is a multifaceted stress corrosion crack. Due to its multifaceted nature, the interaction with the induced eddy current will be highly complex and therefore inversion of the physical geometry would be highly challenging. It is expected that for a simple linear defect, such as a fatigue crack, mixing of datasets would lead to benefits in defect characterisation even if the SNR was adversely affected. This issue is subject to future work and will be investigated by the authors at a later date.

## 4. Conclusions

This paper demonstrates for the first time how eddy current inspection with full image post-processing functions can be robotically deployed, showing a significant step closer to Industry 4.0 applications. Variations in the lift-off of the eddy current array were compensated for by the use of a PI control system and a force–torque sensor ensuring excellent low-noise coupling throughout the inspection. Extensive software infrastructure was developed that allowed for the eddy current data to be post-processed to enhance the generated images and reveal more about the nature of the defects under inspection.

The capability of the eddy current inspection system was demonstrated by inspecting a nuclear canister with a matrix of 16 stress corrosion cracks. Three 180-degree scans were conducted, gathering axial and transversal datasets at four different frequencies simultaneously—250, 300, 400, and 450 kHz—detecting 15/16 stress corrosion cracks. In the resulting data, one defect was highlighted, and various post-processing techniques were employed to increase the image quality. It was shown that, by phase rotation alone, the SNR could be increased by an average of 4.56 decibels. Dataset mixing was also attempted, and it was shown that a selective sum and average could boost the SNR by an average of 1.19 decibels. The multifaceted nature of the stress corrosion crack under inspection created a complex eddy current interaction, making it difficult to invert the physical geometry of the crack. It is expected that for simpler defect geometries a benefit in defect characterisation would be observed through dataset mixing.

This work demonstrated the detection of defects in real-time via eddy current data and showed the ability to further post-process the acquired data to enhance image quality. The benefit of being able to post-process the acquired data in such a manner should not be understated, and it is hoped that similar studies such as this can be used to further develop the post-processing of eddy current data to the standard achieved in ultrasonic NDT.

In future work, the authors plan to improve and progress this study by performing eddy current characterisation on multi-angled known defects and comparing the results to simulated datasets; exploring the use of machine learning to automatically classify and characterise defects; and lastly, exploring the fusion of ultrasonic and eddy current datasets.

## Figures and Tables

**Figure 1 sensors-22-06036-f001:**
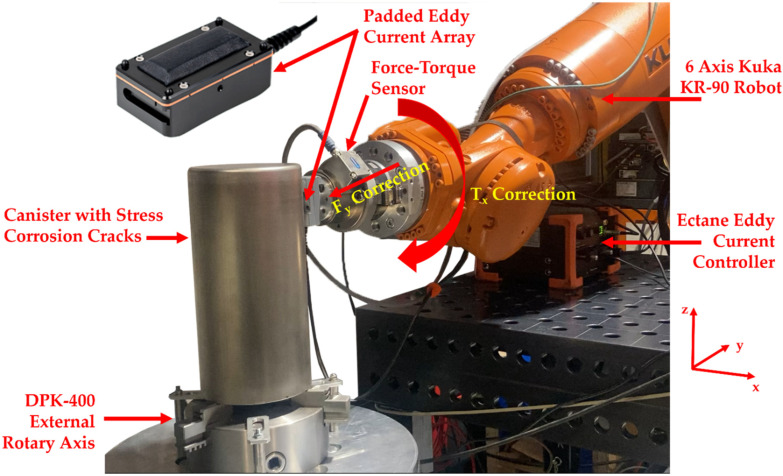
Eddy current inspection hardware.

**Figure 2 sensors-22-06036-f002:**
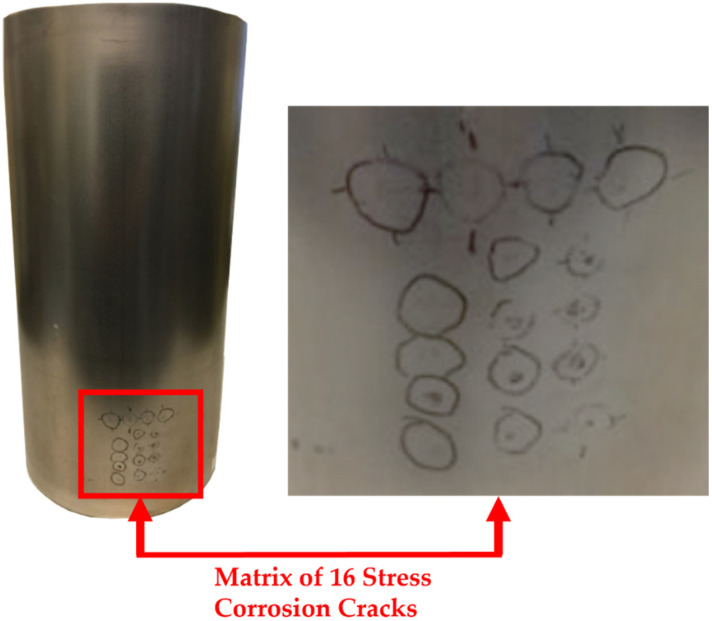
Canisters with a matrix of 16 stress corrosion cracks. Depositions of 5 µL droplets of sea water, 3.03 g/L of MgCl2, 15.2 g/L of MgCl2 and 30.03 g/L of MgCl2 were used to induce the cracks in the top row, left, central and right columns, respectively.

**Figure 3 sensors-22-06036-f003:**
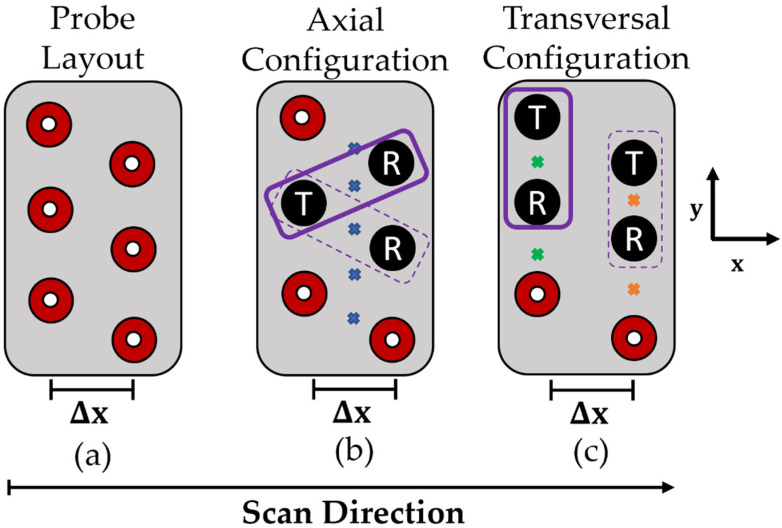
Eddy current array transmit and receive configurations. (**a**) Generic Eddy current array layout with two vertical columns of coils. (**b**) Axial transmit and receive configuration where **x **(**in blue**) corresponds to the transmit/receive pair centres of the excited eddy current channels in the test part. (**c**) Transversal transmit and receive configuration where **x **(**in green**) corresponds to the transmit/receive pair centres of the excited eddy currents in the test part resulting from the first/odd column of coils, and where **x **(**in orange**) corresponds to the transmit/receive pair centres of the excited eddy currents in the test part resulting from the second/even column of coils.

**Figure 4 sensors-22-06036-f004:**
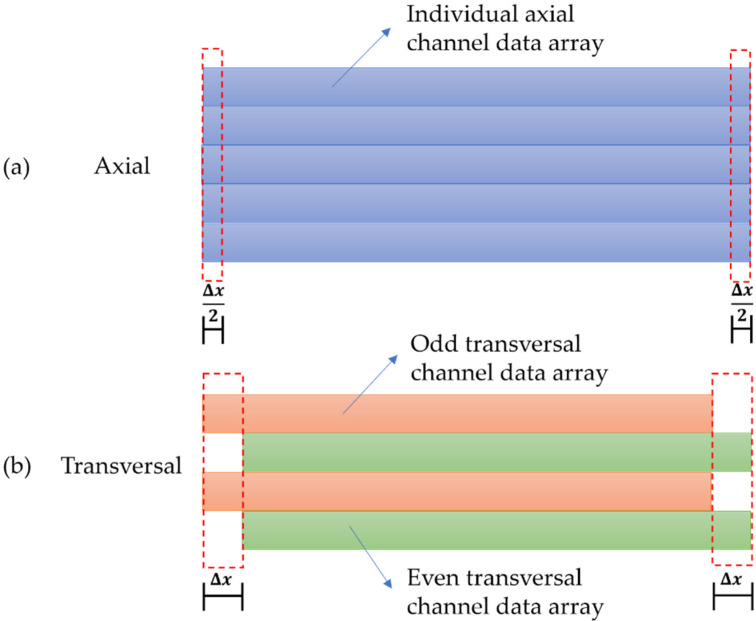
Illustration of complex impedance data positional compensation performed between axial and transversal configurations. (**a**) Axial complex array positional compensation. (**b**) Transversal complex array positional compensation.

**Figure 5 sensors-22-06036-f005:**
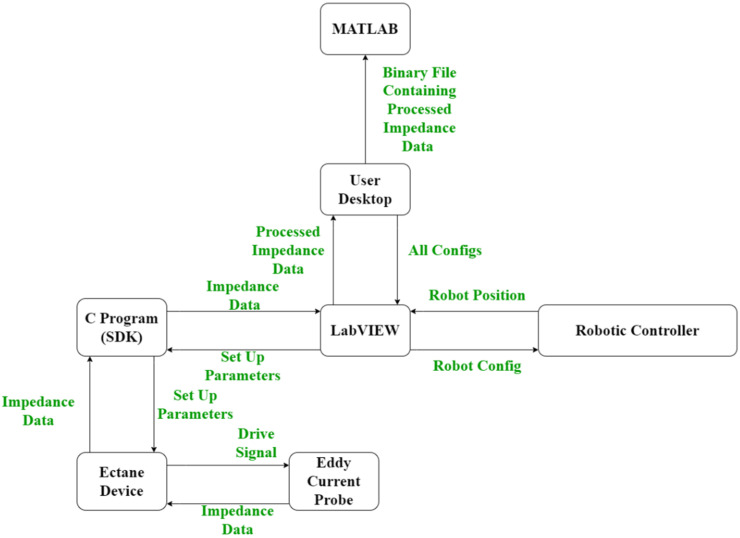
A flow chart showing the data transfer between different software and hardware elements.

**Figure 6 sensors-22-06036-f006:**
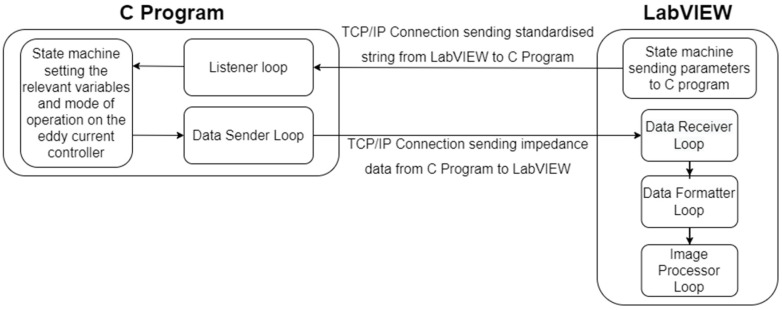
Illustration of the multi-threaded C and LabVIEW programs.

**Figure 7 sensors-22-06036-f007:**
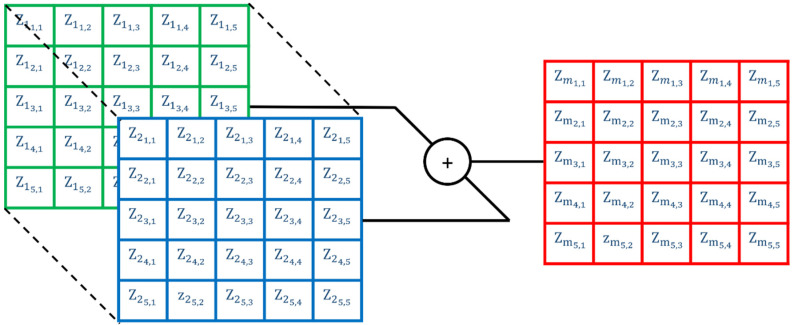
Illustration of mixing datasets $Z_{1}$ and $Z_{2}$ impedance data to make $Z_{m}$ mixed data.

**Figure 8 sensors-22-06036-f008:**
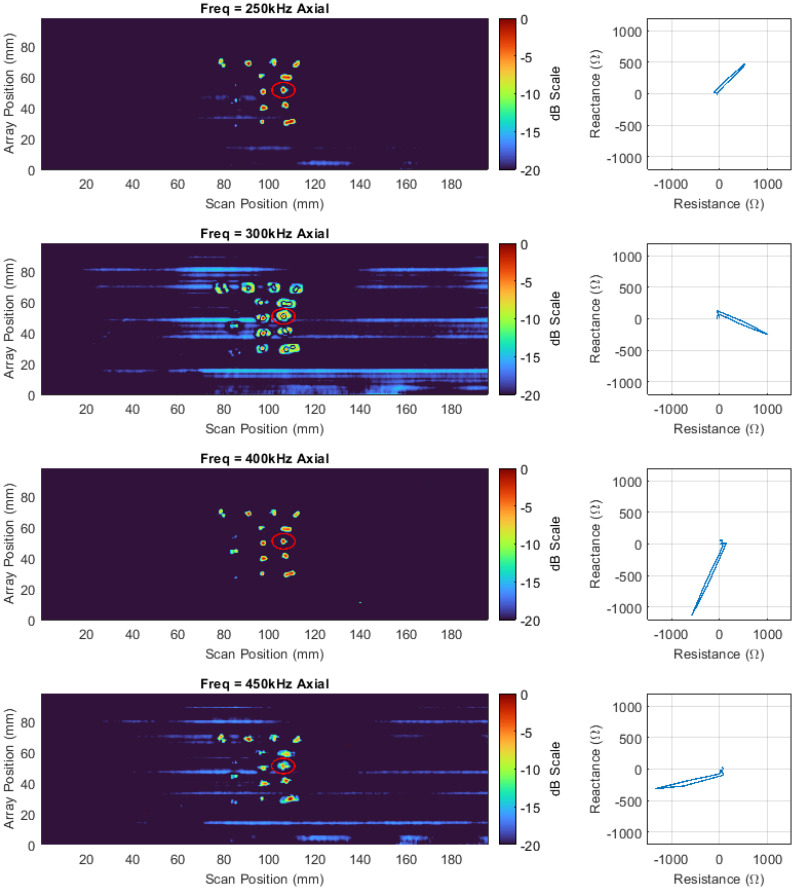
Axial vertical impedance component C−scan images at 250, 300, 400, and 450 kHz on a dB scale alongside impedance plane plots of the response from the highlighted defect.

**Figure 9 sensors-22-06036-f009:**
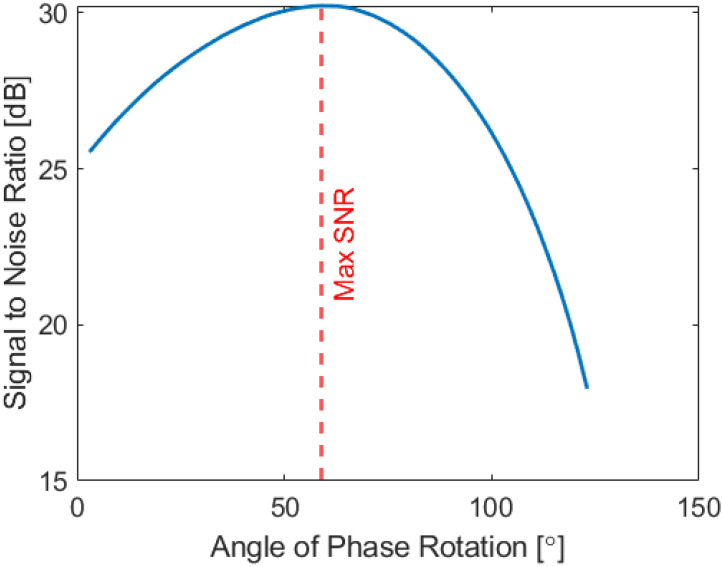
SNR vs. Angle of phase rotation for the axial dataset acquired at 250 kHz.

**Figure 10 sensors-22-06036-f010:**
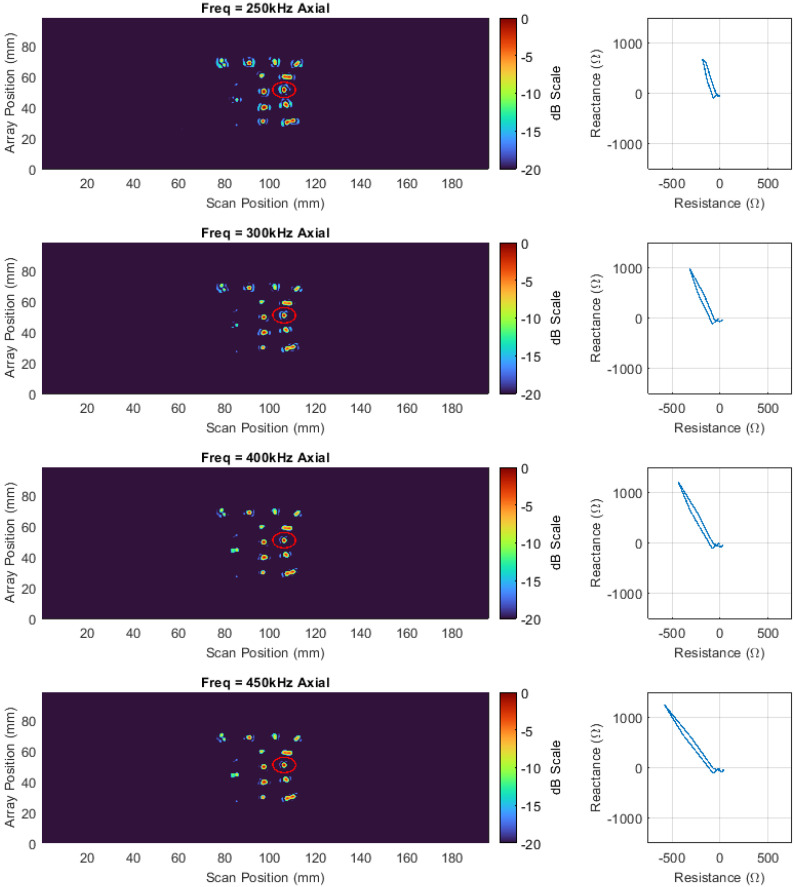
Phase-rotated axial vertical impedance component C−scan images at 250,300,400 and 450 kHz on a dB scale alongside impedance plane plots of the response from the highlighted defect.

**Figure 11 sensors-22-06036-f011:**
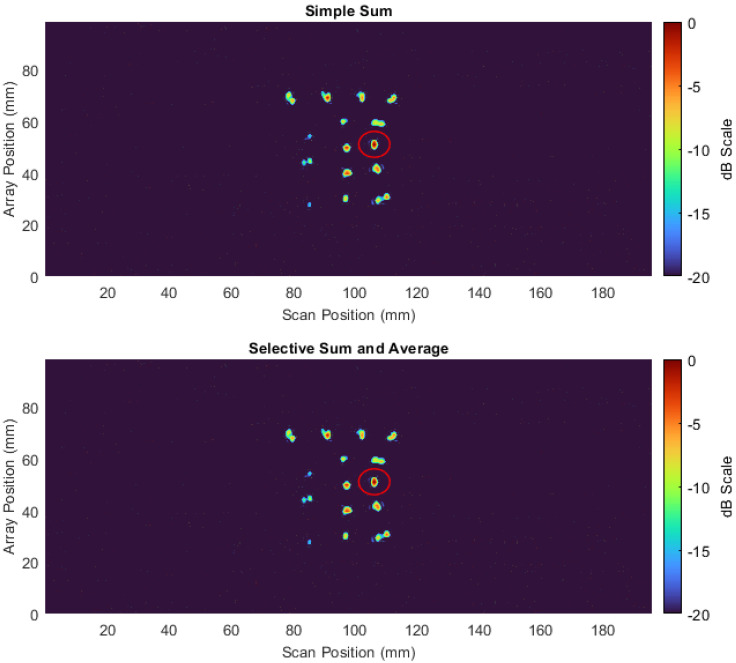
Mixed vertical impedance component C−scan.

**Figure 12 sensors-22-06036-f012:**
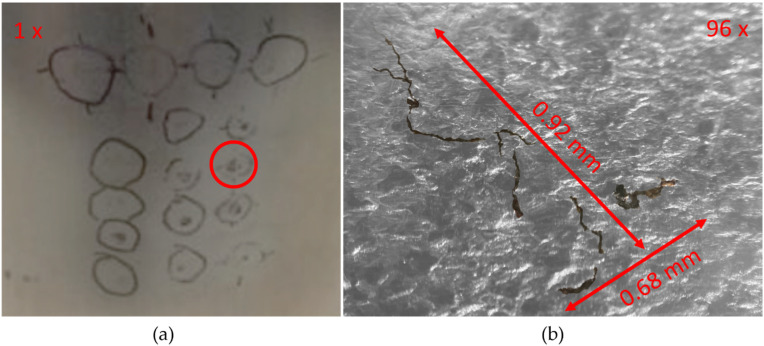
Photo of crack matrix and micrograph (**a**) Photo of crack matrix with the defect of interest highlighted in a red circle. (**b**) Micrograph of the defect of interest at 96× zoom with desaturated background.

**Table 1 sensors-22-06036-t001:** Comparison between previously published robotically deployed eddy current inspections and this work.

	Mackenzie et al., 2009 [25]	Summan et al., 2016 [26]	Morozov et al., 2018 [27]	Zhang et al., 2020 [28]	This Work
Adaptive Motion	**✗**	**✗**	**✗**	**✗**	**✓**
Eddy Current Array	**✗**	**✓**	**✗**	**✓**	**✓**
Image Compensation	**✗**	**✗**	**✗**	**✗**	**✓**

Where **✓** denotes yes and **✗** denotes no.

**Table 2 sensors-22-06036-t002:** SNR Values of original and phase rotated data.

	250 kHz		300 kHz		400 kHz		450 kHz	
	Original SNR [dB]	Phase Rotated SNR [dB]	Original SNR [dB]	Phase Rotated SNR [dB]	Original SNR [dB]	Phase Rotated SNR [dB]	Original SNR [dB]	Phase Rotated SNR [dB]
**Axial**	25.02	30.23	20.19	31.22	29.86	31.27	21.11	31.27
**Transversal**	30.62	32.19	27.43	32.23	31.72	32.28	30.47	32.20

**Table 3 sensors-22-06036-t003:** Mixed Image SNR.

	250 kHz	450 kHz	Mixed Data Simple Sum	Mixed Data Selective Sum
	Phase Rotated SNR [dB]	Phase Rotated SNR [dB]	Phase Rotated SNR [dB]	Phase Rotated SNR [dB]
**Axial**	30.22	31.27	31.85	
**Transversal**	32.19	32.20		32.66

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
