# Peer review of "Automated Real-Time Eddy Current Array Inspection of Nuclear Assets"

_sensors, 2022, doi:10.3390/s22166036_

Round 1

Reviewer 1 Report

The authors present an interesting approach for NDT applications of milli-cracks in a nuclear canister of stainless steel. they use eddy current inspection using 6-axis robotic cell for the scanning process. the research is valuable and provide technological solution to a well-known problem.

The manuscript has a good language and structure that help the readers get to the scientific details easily.

I have few comments:

-The introduction needs to provide more details about the theoretical background and more comparison with different approaches used in comparison to eddy-current inspection.

- what is the error percentage for the calculated crack size in all 3 dimensions?

-how deep can the system identify an embedded crack?

- The authors claim it is a real time evaluation (both in Title and in the manuscript body) while I believe the large amount of processed data would require a long processing time. Can the authors moderate their claim and provide more details about processing time for each algorithm?

- Did the authors consider a weighting function to compensate for the overlap signal amplitude at different scan points? If not, can you please provide info about the calculation error produced with the absence of amplitude weighting (apodization function)?

I think the manuscript would be ready for publication after the authors enhance it considering the mentioned comments.

Reviewer 2 Report

This paper studies the application of array eddy current sensor in Non-Destructive Testing for nuclear material storage tank. The article designs hardware and software systems to realize the functions of automatic data acquisition, real-time data analysis and display. And a new data processing method is proposed to improve the SNR as the optimization goal. This manuscript has the following problems that need to be further improved:

1. There are 16 points marked on the sample, but only the data of one point is used for analysis. Is the obtained conclusion representative?

2. What is the area/range of each test? What is the spatial resolution of the scan?

3. In addition to the variables in the study, other control quantities should also be given specific parameters.

4. It is recommended to supplement the cloud image after data processing in Figure 12 for comparison and analysis with the microscopic image.
